# Research on the Formulation Design of Nano-Oil Displacement Agents Suitable for Xinjiang Jimusaer Shale Oil

**Wei Wang [1], Xianglu Yang [2], Jian Wang [1], Mengjiao Peng [1], Liqiang Ma [3], Mengxiao Xu [2] and Junwei Hou [2,\*]**

[1] Xinjiang Oilfield Company, Karamay 834000, China; 2017592001@cupk.edu.cn (W.W.)
[2] State Key Laboratory of Heavy Oil Processing, China University of Petroleum—Beijing at Karamay, Karamay 834000, China
[3] Shanghai EHS Value Environment Technology Co., Ltd., Shanghai 202150, China
[\*] Correspondence: junweihou@cupk.edu.cn

**Abstract:** In order to improve the recovery efficiency of the Jimusaer tight reservoir in Xinjiang, the nanometer oil displacement agent system suitable for the Jimusaer reservoir was used. In view of the low permeability, high formation temperature, and high salinity characteristics of the prepared water in the Jimusaer tight conglomerate reservoir in Xinjiang, the performance of the nanometer oil displacement agent affecting oil recovery was studied; the study considered interfacial tension, temperature resistance, wetting performance, static oil washing efficiency, and long-term stability. Nanometer oil displacement agent No. 4 had the lowest interfacial tension and could reach the order of $10^{-1}$ mN·m$^{-1}$; it had excellent temperature resistance and the best static oil washing efficiency and stability. Nano-oil displacement agent No. 2 had the best emulsification performance and wettability and also had good stability. By studying the performance and final oil displacement effect of the nano-oil displacement agent, it was found that the key factor affecting the oil displacement effect of this reservoir was the interfacial activity of the nano-oil displacement agent. When the interfacial tension was lower, it produced strong dialysis for oil displacement. The emulsification effect has a negative effect on low-permeability reservoirs, mainly because the fluid produces strong emulsification in low-permeability reservoirs; thus, it can easily block the formation and cause high pressure. An excessive or small contact angle is not conducive to oil displacement. An excessive contact angle means strong hydrophilicity, which can cause a strong Jamin effect in oil-friendly formations. If the contact angle is too small, it has strong lipophilicity and can lead to poor solubility in water. Nano-oil displacement agent No. 4 had the best oil displacement effect, with an oil recovery increase of 7.35%, followed by nanometer oil displacement agent No. 1, with an oil recovery increase of 5.70%. Based on all the performance results, nanometer oil displacement agent No. 4 was more suitable as the oil displacement agent and can be used to enhance oil recovery in the Jimusaer reservoir. This study has laid a foundation for the chemical flooding development of shale oil in the Xinjiang oilfield.

**Keywords:** nano-oil displacement agent; enhanced oil recovery; Jimusaer shale oil; chemical flooding





## 1. Introduction

Nanotechnology plays an important role in the development of modern industry. In the petroleum industry field, nanotechnology is applied in research on improving oil recovery [1]. The role of nanotechnology in improving oil recovery has been a focus of attention in the past decade [2].

Nanotechnology provides a new and promising approach to the improvement of oil recovery in old oil fields. Its application in improving oil recovery can change fluid properties, change the surface wettability of rocks, reduce interfacial tension, increase the flowability of capillary trapped oil, and strengthen sand consolidation [3]. However, the application of nanofluids in the process of improving oil recovery is constrained by various factors, such as the particle size, wettability, mineralization degree, and temperature of the

nanofluids [4–7]. Nanomaterials are a new technology applied to the enhancement of oil recovery in heavy and semi-heavy oil reservoirs, and a new alternative method that can be used to solve the problem of residual oil in heavy and semi-heavy oil reservoirs [8–11]. And many studies have shown that nanofluids improve the process of enhancing oil recovery [12–15]. Nano-oil displacement agents have achieved good results in reducing water content and improving sweep efficiency and oil recovery, and they are especially suitable for tight, low-permeability reservoirs [16–18].

Nano-oil displacement agents are composed of surface-active nanoparticles combined with liquids. Due to the small particle size of nano-oil displacement agents, they can be drilled into narrower channels to expand the wave and volume; they are especially suitable for dense, low-permeability reservoirs. In general, nanoparticles can improve the properties of injected fluid and the interactions between the fluid and the rock, such as interfacial tension, emulsification, wettability, the heat transfer coefficient, etc., to effectively improve oil recovery [19].

Nano-oil displacement agents can form a miscible phase with crude oil; consequently, micelles and a micro-lotion are formed more easily. These micelles can play a role in compatibilizing the oil phase and enhancing the fluidity of the crude oil in the core. Ordinary surfactants have a high loss during the oil displacement process, and they are prone to adsorb on the surface of the core, resulting in poor oil displacement performance. Therefore, their long-term stability is not satisfactory. So, by studying the combination of nanomaterials and surfactants, we aim to achieve lower interfacial tension and stability [20–22].

The tight conglomerate reservoir in Jimusaer, Xinjiang, has drawbacks such as poor physical properties, low permeability, high formation temperature, and high salinity of the liquid preparation water. After investigation, this article evaluates the interfacial tension, emulsification performance, static oil washing, wettability, long-term stability, and dynamic oil washing performance of four nano-oil displacement agents and selects excellent nano-oil displacement agents that are suitable for the dense reservoir of Jimusaer in Xinjiang.

## 2. Materials and Methods

### 2.1. Material and Reagent

The materials used in the experiment are nanometer oil displacement agent No. 1 (nano No. 1), nanometer oil displacement agent No. 2 (nano No. 2), nanometer oil displacement agent No. 3 (nano No. 3), nanometer oil displacement agent No. 4 (nano No. 4), and Xinjiang Jimusaer crude oil. Nanometer oil displacement agent No. 1 is a nano-silicon-based self-penetrating oil displacement agent. Nanometer oil displacement agent No. 2 is a core shell-structured sulfonate polymer compound-coated $Fe_3O_4$ nanocomposite. Nanometer oil displacement agent No. 3 is composed of water, oil, surfactants, isopropanol, and sodium chloride. The surfactant is a combination of polyoxyethylene ether non-ionic surfactants and sulfonated saponins. Nanometer oil displacement agent No.4 is composed of nano-silica, diphenyl ether-based gemini surfactants, isothiazolinone derivatives, enhancers, stabilizers, dodecane, tetradecane, and water.

The water for liquid preparation is Jimusaer 19 water (Ji 19). The water quality analysis results are shown in Table 1. The natural conglomerate core size is $\varphi$ 3.8 cm × 29 cm.

**Table 1.** Analysis results of experimental water quality.

| Name | pH | $CO_3^{2-}$ mg/L | $HCO_3^-$ Mg/L | $OH^-$ mg/L | $Cl^-$ mg/L | $Ca^{2+}$ mg/L | $SO_4^{2-}$ mg/L | $K^+/Na^+$ mg/L | TDS Mg/L | $\rho$ g/cm³ |
|---|---|---|---|---|---|---|---|---|---|---|
| Ji 19 | 7.1 | 0 | 16.89 | 0 | 6604 | 893.6 | 1514 | 3951 | 13,001 | 1.008 |

### 2.2. Experimental Method

(1) Determination method of interfacial tension

The nanometer oil displacement agent solution with different concentration gradients was prepared with the water of Ji 19 from Jimusaer, Xinjiang. The salinity of the water was

13,001 mg·L$^{-1}$. The interfacial tension between the 0.1~0.5% nano-oil displacement agent solution and the crude oil at different temperatures was measured using an interfacial tension meter and a rotating droplet method. The rotational speed was set at 5000 r/min, and the average value was taken after three measurements.

(2)    Determination method of emulsification performance

The solution of the nanometer oil displacement agent with a concentration of 0.3% was prepared with the water of Ji 19 in Xinjiang Jimusaer. Then, the nano-oil displacement agent solution was mixed with the crude oil at a volume ratio of 7:3, and the mixed solution was placed in a water bath for 30 min at 353 K for standby use. Then, the mixed solution was emulsified after the water bath with a high-speed emulsifying machine. The speed of the emulsification machine was 3000 revolutions per minute, and the emulsification time was 5 min. Afterwards, the emulsion was placed in a water bath at 80 °C, the volume of water released was recorded, and the water release rate was calculated.

The water precipitation rate is equal to the percentage of the volume of water precipitation in the total volume of the water phase, and the emulsification performance of the nano-oil displacement agent system is expressed by the water precipitation rate.

(3)    Wettability

Immerse the quartz sheet in dilute hydrochloric acid for 8 h, clean it with deionized water, and then dry it to form an oil film on the surface of the quartz sheet to prepare the lipophilic quartz sheet. After 0.3% of the nanometer oil displacement agent solution and the quartz sheet have been treated at 353 K for a period of time, the contact angle change of the nanometer oil displacement agent solution on the quartz sheet is measured with a contact angle meter, and each nanometer oil displacement agent is measured repeatedly 3 times to take the average value.

(4)    Static oil washing test

The solution of nanometer oil displacement agent with a concentration of 0.3% was prepared with the water of Ji 19 in Xinjiang Jimusaer. Crush, wash, and dry the natural core of Jimusaer, take 150–200 mesh core sand for standby, mix the sand with crude oil, and then place it in a water bath at 353 K for full aging. Then, remove the excess crude oil, obtain the formation oil sand, and then add the nanometer oil displacement agent solution to make it vibrate evenly; finally, obtain the static oil washing efficiency. This experiment used a constant temperature shaker with a rotational speed of 140 rpm.

(5)    Experimental method of nano-lotion flooding

The solution of nanometer oil displacement agent with a concentration of 0.3% was prepared with the water of Ji 19 in Xinjiang Jimusaer. Wash and dry the natural core of Jimusaer, conduct vacuum treatment for 8 h, and then saturate it with formation water; then, measure the pore volume and porosity. Then, age the treated core at 353 K for 8 h to establish the original oil saturation. After water flooding, the water cut reaches 98%, and the recovery factor of the water flooding is obtained. The slug content of the nanometer oil displacement agent solution is 0.7 pv, and the water content reaches 98% after subsequent water flooding. Calculate the increase in the oil recovery rate of the nanometer oil displacement agent system.

*2.3. Analysis Method*

The model and country of the experimental instruments are shown in Table 2.

**Table 2.** Instrument model and country.

| Name | Type | Manufacturer | Place of Origin |
|---|---|---|---|
| Rheometer | MCR301 | Antonpa | Austria |
| Constant temperature water bath | HH-601A | Kanglu | China |
| Microscope | AXIOSKOP 40 | Carl Zeiss Optics | Germany |
| Zeta potentiometer | Nano | Malvern | England |
| Interface tensiometer | TX-500C | Kono Industries Co., Ltd. | America |

## 3. Results

### 3.1. Interfacial Tension

The test results of the interfacial tension between the four different concentrations of nano-oil displacement agent solution and crude oil are shown in Figure 1.

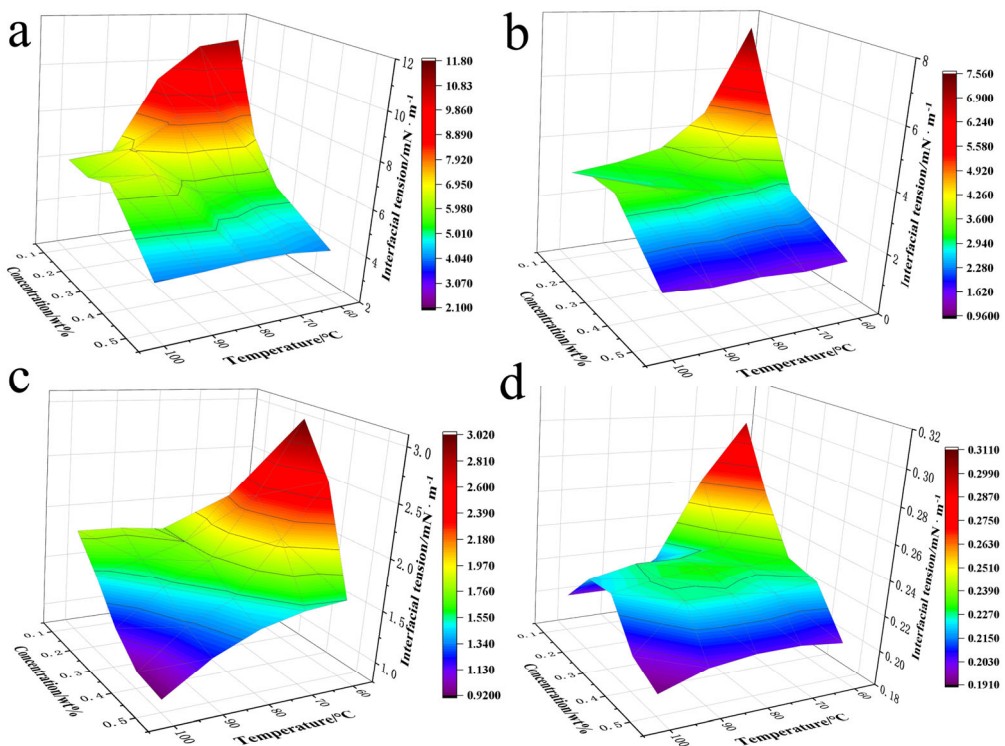

**Figure 1.** Effect of temperature and concentration of nano-oil displacement agent on oil–water interfacial tension: (**a**) nano No. 1; (**b**) nano No. 2; (**c**) nano No. 3; (**d**) nano No. 4.

It is well known that the surfactants reduce the surface tension of water by becoming adsorbed on the liquid–gas interface. Because of the characteristics of Ji 19 water, nano-oil displacement agent 1, nano-oil displacement agent 2, nano-oil displacement agent 3, and nano-oil displacement agent 4 were prepared with Ji 19 water to form 0.1%, 0.2%, 0.3%, 0.4%, and 0.5% solutions. The interfacial tension of the stabilized nano-oil displacement agents at different concentrations and temperatures at 120 min was measured, as shown in Figure 1. At different temperatures, the interfacial tension between the four nano-oil displacement agents and the crude oil showed an overall decreasing trend with increasing concentration. And after the concentration of nano-oil displacement agent reached 0.3%, the effect of the concentration on reducing interfacial tension was relatively small. So, from an economic perspective, it is best to choose a concentration of 0.3% nano-oil displacement agent, which not only saves costs but also maintains a low interfacial tension. Among them, the interfacial tension of nano-oil displacement agent No. 4 was the lowest, and the interfacial tension between the oil and water could reach the order of $10^{-1}$ mN·m$^{-1}$, as shown in Figure 1d. Secondly, the nano-oil displacement agent No. 3 was relatively

low, as shown in Figure 1c. The micelles and micro-lotion formed by the mixing of the nano-oil displacement agent and crude oil can reduce the interfacial energy between oil and water, improve the fluidity of crude oil, and overcome the cohesion between crude oils. Generally speaking, the lower the interfacial tension, the better the dynamic oil displacement effect, and the adhesion work will decrease with the decrease in interfacial tension, thereby improving the oil washing efficiency.

In addition, according to the experimental results in Figure 1, as the temperature increases the interfacial tension between various nano-oil displacement agents and crude oil at low concentrations shows a decreasing trend with increasing temperature. As shown in Figure 1a–c, at higher concentrations, such as concentrations exceeding 0.3%, the interfacial tension between oil and water tends to a stable value at different temperatures, except for nano-oil displacement agent No. 3, indicating that temperature has little effect on its higher concentration of nano-oil displacement agent. For nano-oil displacement agent 3, as shown in Figure 1c, the interfacial tension decreases with the increase in temperature. This indicates that it can only play its best role at higher temperatures, while in general, reservoirs with lower temperatures do not easily achieve optimal results at low temperatures. Therefore, when using nano-oil displacement agents for the Jimusaer reservoir in Xinjiang, nano-oil displacement agents with lower interfacial tension can be selected based on the reservoir temperature, and the viscosity of crude oil and the wetting coefficient of the core change with temperature. As the temperature increases, the thickness of the liquid film on the surface of the core decreases, and the size of the capillary pore throat increases, resulting in a stronger oil displacement effect.

### 3.2. Emulsification Performance

The water precipitation rate of the four nano-oil displacement agent systems with the concentration of 0.3% is shown in Figure 2. They were mixed with the crude oil used in the experiment at 7:3 and heated to 80 °C in a thermostatic water bath. When the oil and water were fully mixed, the volume of water precipitation at different times was recorded. Figure 2 shows the emulsions produced by the different nano-oil displacement agents and crude oil. It can be seen that the emulsions are mainly water-in-oil emulsions. The emulsion size of nano 1 ranges from 20 to 30 microns, with a low density. The emulsion size of nano 2 ranges from 10 to 30 microns, with the highest density and a small amount of oil-in-water emulsion. The emulsion size of nano 3 is above 30 microns and has the lowest density. The emulsion size of nano 4 ranges from 10 to 30 microns and has a high density.

Figure 3a shows the viscosity of crude oil and the viscosity of the emulsion after emulsification with nano-oil displacement agents. It can be seen that the initial viscosity of crude oil is low, only 6 mPa·S. The viscosity after emulsification with the No. 1 oil displacement agent reaches 13 mPa·S. The viscosity after emulsification with the No. 2 oil displacement agent is the highest, reaching 21 mPa·S. The viscosities of the emulsified oil displacement agents 3 and 4 are 8 mPa·S and 11 mPa·S, respectively. Figure 3b shows the emulsification performance of different oil displacement systems with a concentration of 0.3%. After the emulsification of crude oil and the nanometer oil displacement agent system, a crude oil emulsion is formed, which can increase the viscosity of the system and bring the crude oil out of the formation. The particle size of the nano-micro-lotion reached the nanometer level and was able to smoothly enter the micropores in the tight reservoirs, expand the swept volume, and further improve oil recovery. The water evolution rate of nano-oil displacement agent No. 2 was kept at a low level, indicating that its emulsification performance was the best. The emulsification performances of nanometer oil displacement agent No. 1 and nanometer oil displacement agent No. 4 were average, and the water evolution rates at 0.25 h were 70% and 83%, respectively. The emulsification performance of nanometer oil displacement agent No. 3 was the worst; it was completely demulsified and dehydrated at 1h, and the water separation rate reached 100%.

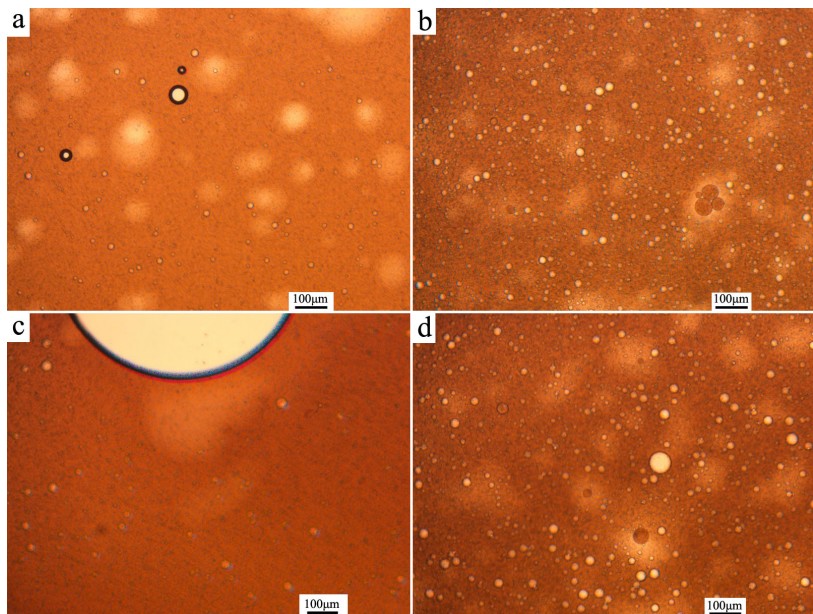

**Figure 2.** Emulsification photo of different oil displacement systems with concentration of 0.3%: (**a**) nano No. 1; (**b**) nano No. 2; (**c**) nano No. 3; (**d**) nano No. 4.

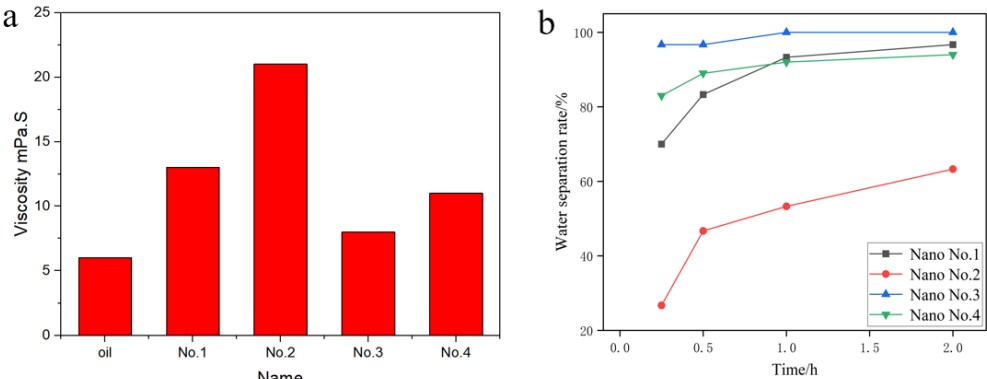

**Figure 3.** Emulsification performance of different oil displacement systems with concentration of 0.3%: (**a**) viscosity; (**b**) demulsification rate.

### 3.3. Static Oil Washing Test

The static oil washing efficiency of four nano-oil displacement agents and clean water is shown in Figure 4. It can be seen from Figure 3 that the static oil washing efficiency of the four nano-oil displacement agents gradually increases with time. The increase rate is large in the first 2 h, and the static oil washing efficiency tends to be stable after more than 2 h. Among them, nanometer oil displacement agent No. 4 has the best static oil displacement effect and can reach about 60%. The static oil washing efficiency of nano-oil displacement agents 1~3, after stabilization, is very close, about 50%. In addition, the oil washing efficiency of the four nano-oil displacement agents is higher than that of clean water, and the highest oil washing efficiency of clean water is 35%. A nano-oil displacement agent peels the crude oil on the core surface into small particles of oil droplets, making it easier to wash the crude oil out of the rock pores, thus improving the oil washing efficiency. Therefore, it is reasonable to use nanometer oil displacement agent No. 4 to improve the static oil washing efficiency and to further investigate the oil displacement efficiency.

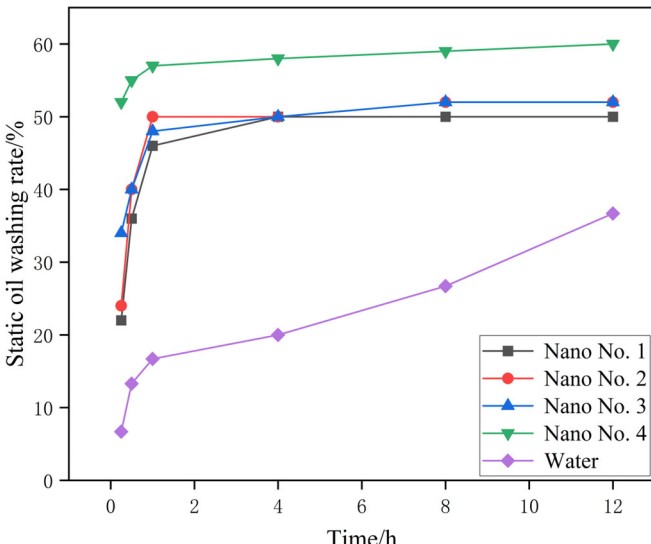

**Figure 4.** Static oil washing rate of different oil displacement systems with 0.3% concentration.

### 3.4. Wettability

The wettability of four kinds of nanometer oil displacement agents was measured. The results are shown in Figure 4.

Contact angle is an important indicator for measuring the wettability reversal ability of surfactants. Nano-oil displacement agents, due to the large specific surface area of the nano-oil displacement fluid, are conducive to the diffusion of the fluid on the surface of the medium, which means it has good wettability. As shown in Figure 5, nano-oil displacement agent No. 1 and nano-oil displacement agent No. 4 have large contact angles, and the contact angle decreases slowly with the increase in time. The contact angle of nano-oil displacement agent No. 3 decreases rapidly with the increase in time. The contact angle of nano-oil displacement agent No. 2 is the smallest and is only about 20°. The smaller the contact angle, the stronger the hydrophilicity of the core surface; thus, the oil recovery factor is improved. The wetting reversal effect of the nanometer oil displacement agent is obvious. The nanometer oil displacement agent can transform the oil-wet surface into a strong water-wet surface and can then convert the resistance of the capillary into the driving force of oil displacement. The change in wettability can also reduce the adhesion work of the crude oil in the formation, making it easier for the crude oil to elute from the rock surface.

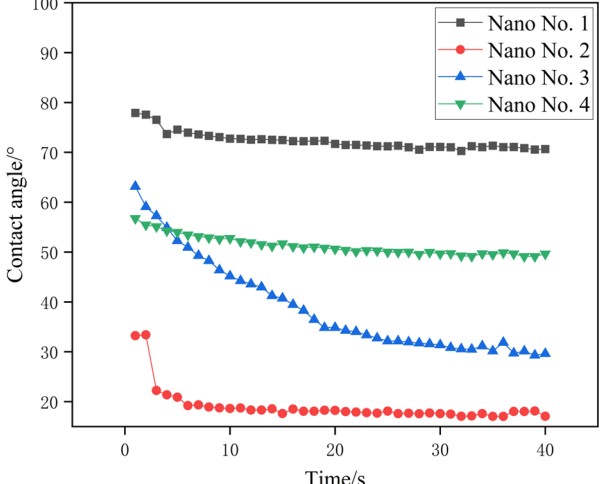

**Figure 5.** Comparison of contact angles of different nano-oil displacement agents with concentration of 0.3%.

### 3.5. Long-Term Stability Test

Under ideal conditions, the stability of the nano-oil displacement agent does not change with time, but under actual reservoir conditions, the stability changes under the influence of water salinity and temperature. The stability of the nanometer oil displacement agent within 60 days was studied using four kinds of nanometer oil displacement agents with a concentration of 0.3% under reservoir conditions, and Figure 4 was drawn according to the relationship between interfacial tension and time. It can be seen from Figure 6 that nano-oil displacement agent No. 2 and nano-oil displacement agent 4 can maintain stability within 60 days and that the interfacial tension is at a low level. With the change in time, the interfacial tension of nanometer oil displacement agent No. 3 increased suddenly from 5 to 15 days and then decreased to the initial level. However, the interfacial tension of nanometer oil displacement agent No. 1 increased greatly with the increase in time, indicating that its stability was poor. It can be seen from the analysis that under the actual reservoir conditions nanometer oil displacement agent No. 2 and nanometer oil displacement agent No. 4 can play a relatively stable role.

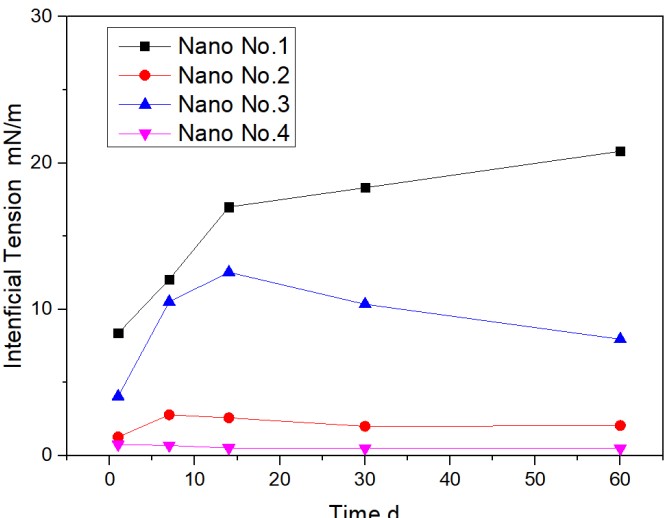

**Figure 6.** Long-term stability.

### 3.6. Dynamic Oil Displacement

In order to investigate the ability of different types of nanometer oil displacement agent systems to improve the oil displacement effect, the nanometer oil displacement agent systems with a concentration of 0.3% were classified according to the main performance evaluation results, such as interfacial tension, emulsification performance, and wettability, as shown in Table 3. They were divided into three types: the medium interfacial activity, weak emulsification, weak wettability type; the medium interfacial activity, strong emulsification, strong wettability type; the medium interfacial activity, non-emulsification, medium wettability type; and the low interfacial, active weak emulsification, medium wettability type. The classification according to the above types is more conducive to the analysis of experimental results.

**Table 3.** Classification of basic performance of four nano-oil displacement systems with concentration of 0.3%.

| Oil Displacement System | Interfacial Tension (mN·m$^{-1}$) | 0.25 h Water Evolution Rate (%) | Contact Angle (°) | Enhanced Oil Recovery (%) |
| --- | --- | --- | --- | --- |
| Nano No. 1 | 5.62 | 70 | 70.66 | 3.98 |
| Nano No. 2 | 3.11 | 26.7 | 17.06 | 4.51 |
| Nano No. 3 | 1.72 | 96.7 | 29.62 | 5.70 |
| Nano No. 4 | 0.231 | 83 | 49.63 | 7.35 |

Figures 7–10 show the dynamic oil displacement curves of four nano-oil displacement agents, respectively. The horizontal axis represents the injection amount, the left vertical axis represents the water content, and the right axis represents the recovery rate. The core permeability is the average permeability of the study area (10 mD), and the surfactant slug volume is 0.7 PV. The core size used is shown in Table 3. It can be seen from the figure that when the injection volume is small, it is water driven, with a low water cut and a large increase in oil recovery. Then, when the water cut increases to about 98%, the nanometer oil displacement agent is used for oil displacement, the water cut decreases, and the oil recovery continues to increase. The oil displacement effect of the nanometer oil displacement agent with middle interfacial activity, strong emulsification, and strong wettability and middle interfacial activity, non-emulsification, and medium wettability is general, and the water cut after adding the nanometer oil displacement agent is not obvious; the enhanced oil recoveries are only 3.98% and 4.51%. This shows that for the Jimusaer tight reservoir in Xinjiang improving oil recovery requires a nanometer oil displacement agent with moderate emulsification performance and that strong emulsification or non-emulsification has a limited effect on improving the oil displacement effect of the Jimusaer tight reservoir.

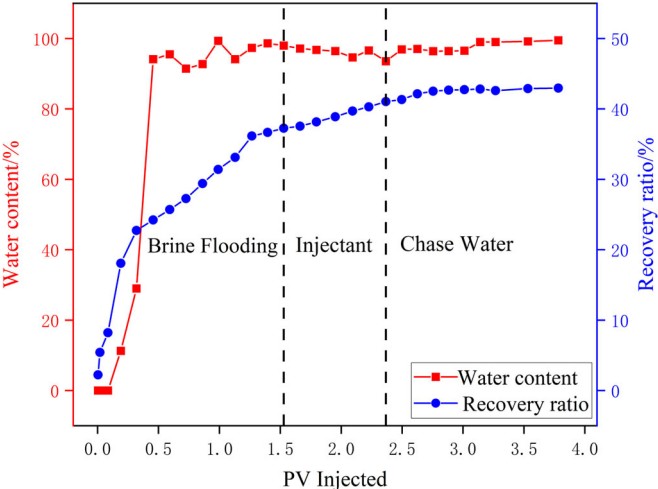

**Figure 7.** Core oil displacement results of nano No. 1.

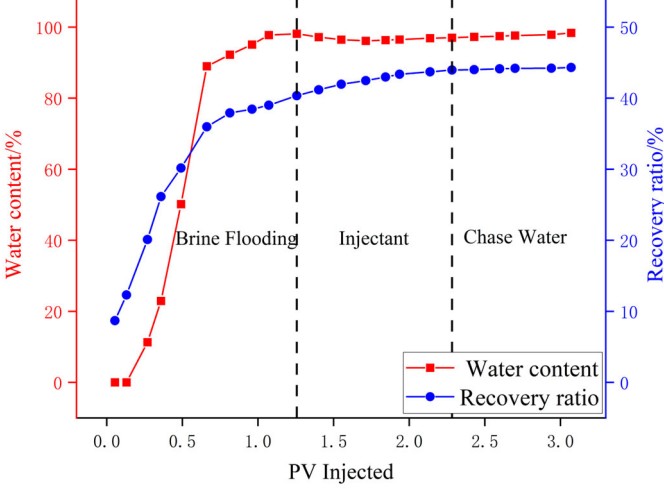

**Figure 8.** Core oil displacement results of nano No. 2.

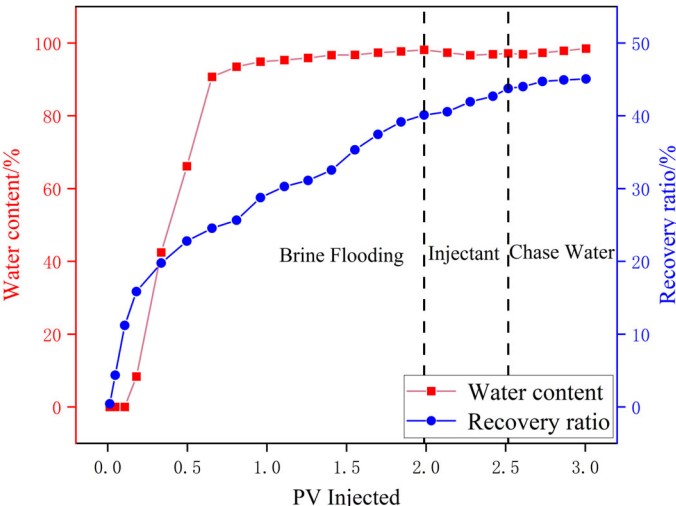

**Figure 9.** Core oil displacement results of nano No. 3.

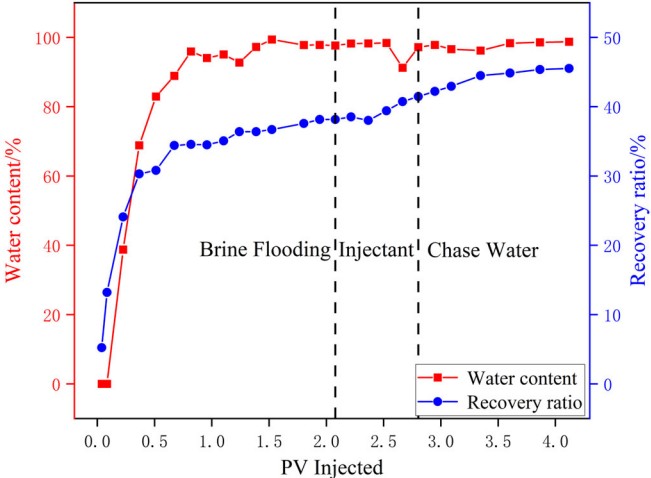

**Figure 10.** Core oil displacement results of nano No. 4.

Compared with nano No. 1, nano No. 4 has 1.8 times higher oil recovery efficiency and 7.35% oil recovery efficiency, while nanometer oil displacement agent No. 4 has the lowest interfacial activity and moderate wettability; this indicates that the interfacial activity of the nanometer oil displacement agent makes the highest contribution to the improvement of oil recovery for this reservoir and that it is necessary for the nanometer oil displacement agent to have certain wettability because a proper change in wettability can reduce adhesion work. Thus, the crude oil will be washed out, and the oil recovery will be improved.

Table 3 shows the core parameters with the oil displacement test data. Nano-oil displacement agent No. 4 has the best oil displacement effect, with an oil recovery increase of 7.35%. The second is nanometer oil displacement agent No. 1, with an increase in the oil recovery of 5.70%. The oil displacement effect of nano-oil displacement agent No. 2 and nano-oil displacement agent No. 3 is average at only 3.98% and 4.51%. According to the comprehensive evaluation results, nanometer oil displacement agent No. 4 is more suitable for use as the oil displacement agent for enhancing oil recovery in Jimusaer reservoir, and nanometer oil displacement agent No. 4 has been put into use in Block Ji-45.

## 4. Discussion

Compared with the four kinds of nanometer oil displacement agents, nano No. 4 has the lowest interfacial tension; it can reach the order of $10^{-1}$ mN·m$^{-1}$ and can maintain stable interfacial tension at high temperature, with excellent temperature resistance. Nano

No. 2 has the best emulsification performance, followed by nano No. 4 and nano No. 1, while nano-oil displacement agent No. 3 has almost no emulsification ability. Nano No. 4 has the best static oil washing efficiency, and nano-oil displacement agent No. 2 has the best wettability. Under the actual reservoir conditions, nano-oil displacement agent No. 2 and nano-oil displacement agent No.4 have the best stability.

## 5. Conclusions

In view of the low permeability, high formation temperature, and high salinity characteristics of the prepared water in the Xinjiang Jimusaer tight conglomerate reservoir, the dynamic oil displacement experiment was carried out; the results showed that nanometer oil displacement agent No. 4, which was a low interfacial activity, weak emulsification, and wettability type, had the best oil displacement effect, indicating that the interface performance of the nanometer oil displacement agent played a major role in improving the oil displacement effect for Xinjiang Jimusaer reservoir exploitation and that it was necessary for the nanometer oil displacement agent to have certain emulsification ability and good wettability. According to the dynamic oil displacement experiment, the oil displacement effect of nanometer oil displacement agent No. 4 is the best, with an increase in oil recovery of 7.35%, followed by nanometer oil displacement agent No. 1, with an increase in oil recovery of 5.70%. Based on all the performance results, the nanometer oil displacement agent No. 4 is more suitable for use as the oil displacement agent to enhance oil recovery in Jimusaer Reservoir, and the nanometer oil displacement agent No. 4 has been put into use in Block Ji-45.

**Author Contributions:** Conceptualization, W.W.; data curation, X.Y.; funding acquisition, J.H.; investigation, J.W.; methodology, M.P. and L.M.; software, M.X.; supervision, J.H.; visualization, W.W.; writing—review and editing, X.Y. All authors have read and agreed to the published version of the manuscript.

**Funding:** This research was funded by [Tianshan Young Scholars] grant number [2018Q031] and [Educational Foundation of Xinjiang Province] grant number [XJEDU2018Y060].

**Data Availability Statement:** The authors confirm that the data supporting the findings of this study are available within the article.

**Conflicts of Interest:** The authors declare no conflict of interest.

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
