# Peer review of "Research on the Formulation Design of Nano-Oil Displacement Agents Suitable for Xinjiang Jimusaer Shale Oil"

_processes, doi:10.3390/pr11092610_

Round 1
Reviewer 1 Report
This article, studying the performance of a series ofnanometer oil displacement agents affecting oil recovery, including interfacial tension, temperature resistance, wetting performance, static oil washing efficiency and long-term stability. Based on all the performance results, the nanometer oil displacement agent No. 4 is more suitable as the oil displacement agent used for enhancing oil recovery in Jimusaer reservoir.
1. This article lacks of innovation which only studied the performance of a series of nanometer oil displacement agents affecting oil recovery. 2. Jimusaer tight conglomerate reservoir in Xinjiang hasthe characteristics of low permeability, high formation temperature and high salinity of the prepared water,whether salt tolerance is considered in the design of the nano oil displacement agent? 3. “Oil displacement” is repeated with “Nano oil displacement agent” in Keywords. 4. The experimental method used in this paper is simple and innovative. 5. What's the formula for the nano oil displacement agents? only Ji 19 water and nanomaterials? What nanomaterials are used? What is the dispersion stability of nanomaterials? 6. What is the actual temperature and pressure of the reservoir? How do laboratory experiments ensure consistency with field reservoir conditions? 7. “when the water cut increases to about 98%, nanometer oil displacement agent is used for oil displacement, and the water cut decreases and the oil recovery continues to increase” was mentioned in the text, but the chart does not show this trend. 8. Nano emulsion Flooding are mixed with the crude oil at 7:3, but there was almost no change in water content after injection of nanometer oil displacement agent,whether this phenomenon is reasonable? 9. The conclusion is too simple and does not play the role of summarizing the full text.Author Response
- This article lacks of innovation which only studied the performance of a series of nanometer oil displacement agents affecting oil recovery.
Reply: Thank the reviewers for their suggestions, the innovation has been added to the revised paper.
- Jimusaer tight conglomerate reservoir in Xinjiang has the characteristics of low permeability, high formation temperature and high salinity of the prepared water, whether salt tolerance is considered in the design of the nano oil displacement agent?
Reply: Thank the reviewers for their suggestions, the temperature is 80℃,and the salt concentration is 13000mg/L. The experimental conditions were all conducted at a temperature of 80 ℃ and a mineralization degree of 13000mg/L.
- “Oil displacement” is repeated with “Nano oil displacement agent” in Keywords.
Reply: Thank the reviewers for their suggestions, the new keyword has been added to the revised paper.
- The experimental method used in this paper is simple and innovative.
Reply: Thank the reviewers for their suggestions, the experimental method has been redefined in the revised paper.
- What's the formula for the nano oil displacement agents? only Ji 19 water and nanomaterials? What nanomaterials are used? What is the dispersion stability of nanomaterials?
Reply: Thank the reviewers for their suggestions, The permeability of shale reservoir is very low, and polymer cannot be used, so our formula mainly uses nano oil displacement agent and water for chemical flooding.
- What is the actual temperature and pressure of the reservoir? How do laboratory experiments ensure consistency with field reservoir conditions?
Reply: Thank the reviewers for their suggestions, the temperature is 80℃.
- “When the water cut increases to about 98%, nanometer oil displacement agent is used for oil displacement, and the water cut decreases and the oil recovery continues to increase” was mentioned in the text, but the chart does not show this trend.
Reply: Thank the reviewers for their suggestions. From Figure 8, it can be seen that when the injection volume reaches 2.7 PV, there is a significant decrease in water content and an increase in oil recovery. However, for Shale oil, the water cut decreases for a short time and rebounds quickly.
- Nano emulsion Flooding are mixed with the crude oil at 7:3, but there was almost no change in water content after injection of nanometer oil displacement agent, whether this phenomenon is reasonable?
Reply: Thank the reviewers for their suggestions. In the demulsification experiment, we used crude oil and chemical agents in a ratio of 7:3 for emulsification, and then observed the trend of demulsification. The oil-water ratio in the dynamic oil displacement experiment is not 7:3, and the strength and emulsification method of emulsification and hand emulsification during the oil displacement process are different, so the results are also different.
- The conclusion is too simple and does not play the role of summarizing the full text.
Reply: Thank the reviewers for their suggestions, the conclusion has been redefined in the revised paper.

Reviewer 2 Report
This manuscript characterizes four nano oil displacement agents, and the experimental work is complete. The findings of this research hold significant theoretical implications. Please find below a list of detailed comments and questions.
(1) In section 2.2, Please add complete emulsification conditions, such as emulsification time, etc.
(2) The significance of wettability determination by quartz stone is less clear.
(3) Why was WOR selected as 7:3?
(4) In section 2.2, the static oil washing test was vague. What kind of vibration is it? Hands or machine?
(5) It might be clearer to change Figure 1 to 2D.
(6) Why is aging in the constant temperature water bath and not in the oven?
(7) When measuring the interfacial tension, the aging time is too long, and the emulsion have separated. How to measure the interfacial tension?
(7) The recovery rate in the oil displacement experiments was almost the same. It is suggested to supplement several groups of controlled experiments.
(8) Additionally, there are some mistakes in the writing, and textual logic need to revisit.
Moderate editing of English language required
Author Response
- In section 2.2, Please add complete emulsification conditions, such as emulsification time, etc.
Reply: Thank the reviewers for their suggestions, the emulsification conditions has been redefined in the revised paper.
- The significance of wettability determination by quartz stone is less clear.
Reply: Thank the reviewers for their suggestions, The purpose of treating quartz is to change its wettability from hydrophilicity to lipophilicity.
- Why was WOR selected as 7:3?
Reply: Thank the reviewers for their suggestions. According to the book "Petroleum Emulsion", the ratio of crude oil to water emulsified in remote well areas is mainly 7:3. the new references has been redefined in the revised paper.
- In section 2.2, the static oil washing test was vague. What kind of vibration is it? Hands or machine?
Reply: Thank the reviewers for their suggestions. This experiment used a constant temperature shaker with a rotational speed of 140rpm.
- It might be clearer to change Figure 1 to 2D.
- Why is aging in the constant temperature water bath and not in the oven?
Reply: Thank the reviewers for their suggestions. The temperature control of a water bath is more accurate than an oven
- When measuring the interfacial tension, the aging time is too long, and the emulsion have separated. How to measure the interfacial tension?
Reply: Thank the reviewers for their suggestions. The interfacial tension is measured using an interfacial tension meter, while emulsification is evaluated using water separation rate. It is measured separately.
- The recovery rate in the oil displacement experiments was almost the same. It is suggested to supplement several groups of controlled experiments.
Reply: Thank the reviewers for their suggestions. The data on improving oil recovery has been added to Table 3, and subsequent experiments will be presented in a new paper.
- Additionally, there are some mistakes in the writing, and textual logic need to revisit.
Reply: Thank the reviewers for their suggestions. The text has been modified

Reviewer 3 Report
The process of heavy oil upgrading and recovery using nanoparticles as adsorbent/catalysts is quite new and challenging chemical process. The title of the manuscript is “Research on the formulation design of nano oil displacement agents suitable for Xinjiang Jimusaer shale oil”. What are the components of nanometer oil displacement agents? Were they prepared by nanomaterials and surfactants? It would be important to add more clarifications about the components of nanometer oil displacement agents?
Besides, before further processing of the manuscript, some issues should also be addressed in more depth and details:
Q1: Many sentences are informal and difficult to understand, for instance, “This study provides ideas for the treatment of emulsion in the later stage of oilfield development.” “As an achievement of nanotechnology, the significance of nanofluids in industrial applications is beyond doubt.” “Nanotechnology mainly involves adding nano particles to known liquids to form oil displacement agents.” Line 151-152, 162-163. Line 216-218
Q2: Line 17 “1 mN∙m-1” should be “1 mN∙m-1”. Line 79, Table 1, “mg.L” should be “mg/L” Line 84, “13001mg∙L-1” should be “13001mg∙L-1” . Line 98, W should be in upper case. Line 127, it should be “interfacial tension”.
Q3: Line 54, what does “improving interfacial tension” mean?
Q4: Line 131, it should be “Figure 1.” and the axes are not clear.
Q5: Line 181-183, what is the size of the emulision droplets? Do crude oil emulsions increase or decrease the viscosity of the system?
Q6: Line 246, What is “Nanoemulsion” and how does it prepare?
Q7 : The Discussion section sounds more like a short Conclusion overview and should be edited.
Extensive editing of English language required
Author Response
- Many sentences are informal and difficult to understand, for instance, “This study provides ideas for the treatment of emulsion in the later stage of oilfield development.” “As an achievement of nanotechnology, the significance of nanofluids in industrial applications is beyond doubt.” “Nanotechnology mainly involves adding nano particles to known liquids to form oil displacement agents.” Line 151-152, 162-163. Line 216-218
Reply:Thank the reviewers for their suggestions, these sentences have been redefined in the revised paper.
- Line 17 “1 mN∙m-1” should be “1 mN∙m-1”. Line 79, Table 1, “L” should be “mg/L” Line 84, “13001mg∙L-1” should be “13001mg∙L-1” . Line 98, W should be in upper case. Line 127, it should be “interfacial tension”.
Reply: Thank the reviewers for their suggestions. This error has been corrected in the revised paper.
Q3: Line 54, what does “improving interfacial tension” mean?
Reply: Thank the reviewers for their suggestions. This error has been corrected in the revised paper.
Q4: Line 131, it should be “Figure 1.” and the axes are not clear.
Reply: Thank the reviewers for their suggestions. This error has been corrected in the revised paper.
Q5: Line 181-183, what is the size of the emulision droplets? Do crude oil emulsions increase or decrease the viscosity of the system?
Reply: Thank the reviewers for their suggestions. The size of the emulision droplets has been added in the revised paper. The Crude oil emulsions increase the viscosity of the system.
Q6: Line 246, What is “Nanoemulsion” and how does it prepare?
Reply: Thank the reviewers for their suggestions. This error has been corrected in the revised paper.
Q7 : The Discussion section sounds more like a short Conclusion overview and should be edited.
Reply: Thank the reviewers for their suggestions. This error has been corrected in the revised paper.

Reviewer 4 Report
The review of the manuscript entitled: “Research on the formulation design of nano oil displacement agents suitable for Xinjiang Jimusaer shale oil”. The authors completed very interesting work. The work is in the scope of the journal. By responding to the following comments, the work can be ready for publication:
1. In the abstract, writing “displacement agent No. 4” or “Nano oil displacement agent No. 2” is not suitable. Please rewrite it.
2. The mechanism of EOR in the abstract should be added. Why it occurred.
3. Introduction section should be extended. Also, in this section, please consider the other technologies for the increase in oil production such as asphaltene control. For this purpose, the next reference can be used in the revision stage: Khormali, A. (2022). Effect of water cut on the performance of an asphaltene inhibitor package: experimental and modeling analysis. Petroleum Science and Technology, 40(23), 2890-2906.
4. By which method the ion content of water was determined?
5. Please add pictures of the used apparatuses.
6. What was the kind of emulsion? Water in oil or oil in water?
7. It is recommended to study the dynamic conditions for the experiments.
Author Response
- In the abstract, writing “displacement agent No. 4” or “Nano oil displacement agent No. 2” is not suitable. Please rewrite it.
Reply: Thank the reviewers for their suggestions. This error has been corrected in the revised paper.
- The mechanism of EOR in the abstract should be added. Why it occurred.
Reply: Thank the reviewers for their suggestions. This mechanism of EOR has been added in the revised paper.
- Introduction section should be extended. Also, in this section, please consider the other technologies for the increase in oil production such as asphaltene control. For this purpose, the next reference can be used in the revision stage: Khormali, A. (2022). Effect of water cut on the performance of an asphaltene inhibitor package: experimental and modeling analysis. Petroleum Science and Technology, 40(23), 2890-2906.
Reply: Thank the reviewers for their suggestions. The new reference [22] has been added in the revised paper.
- By which method the ion content of water was determined?
Reply: Thank the reviewers for their suggestions. The titration method is used to measure ion content
- Please add pictures of the used apparatuses.
Reply: Thank the reviewers for their suggestions. However, there are already 9 images in the paper, and the device images cannot be added anymore.
- What was the kind of emulsion? Water in oil or oil in water?
Reply: Thank the reviewers for their suggestions. The kind of emulsion has been corrected in the revised paper.
- It is recommended to study the dynamic conditions for the experiments.
Reply: Thank the reviewers for their suggestions. The detailed dynamic conditions will be discussed in the next paper.

Reviewer 5 Report
The manuscript present no identification data about the materials used for analysis so the results can't be properly analysed. Thus, the reader can't check whether the conclusions are supported by the results.
Author Response
Thank you, reviewer
Reviewer 6 Report
Dear Authors,
The paper entitled: Research on the formulation design of nano oil displacement agents suitable for Xinjiang Jimusaer shale oil can be interesting, but in my opinion not suitable for publication in Processes.
1. The first and main problem of this paper is the fact that no one else in the world will ever repeat the results obtained in this paper.Thus, no one in the world will ever check whether these studies are correct or not.
The composition of the “mysterious” four nano oil displacing agents (Agent no.1-4) is not completely known.
The author, reading the work, wonders if they exist.
Therefore, unfortunately, I propose to reject this work. If the composition of these compounds is, for example,
in the process of patenting, then publication should be deferred until the patent is obtained.
2. Additional questions to be clarified are:
Is figure 5 (and table 3) related to section 3.5 Interface Tension (should be interfacial tension)? It is not clear to the reader. for example, in Figure 5, the interfacial tension for agent 1 (at 3% concentration - line 234) at time 0 is about 15 mN/m. Looking at Figure 1a, readers do not see Agent 1 (or any other agent) reaching this value.
Author Response
- The first and main problem of this paper is the fact that no one else in the world will ever repeat the results obtained in this paper.Thus, no one in the world will ever check whether these studies are correct or not.The composition of the “mysterious” four nano oil displacing agents (Agent no.1-4) is not completely known.The author, reading the work, wonders if they exist.
Therefore, unfortunately, I propose to reject this work. If the composition of these compounds is, for example,in the process of patenting, then publication should be deferred until the patent is obtained.
Reply: Thank the reviewers for their suggestions. We will improve the research quality in future papers.
- Additional questions to be clarified are:
Is figure 5 (and table 3) related to section 3.5 Interface Tension (should be interfacial tension)? It is not clear to the reader. for example, in Figure 5, the interfacial tension for agent 1 (at 3% concentration - line 234) at time 0 is about 15 mN/m. Looking at Figure 1a, readers do not see Agent 1 (or any other agent) reaching this value.
Reply: Thank the reviewers for their suggestions. This is our mistake; we have measured the data in the revised paper.
Round 2
Reviewer 3 Report
Minor editing of English language and text editing required.
Minor editing of English language and text editing required.
Author Response
Thank the reviewers for their suggestions, the The minor English errors have been corrected in the revised paper.
Reviewer 4 Report
the work was revised based on the comments. Thus, it is recommended for publication.
Author Response
Thank the reviewers for their suggestions.
Reviewer 5 Report
The paper has been duly revised. It remains a mystery to me what nanometer oil displacement agent no.1, no 2, etc are. At least the authors should comment on how these agents differ from each other. For this reason I have to reject this work.
Author Response
Thank the reviewers for their suggestions. The nanometer oil displacement agent no.1 is Nano silicon based self-penetrating oil displacement agent. The nanometer oil displacement agent no.2 is the core shell structured sulfonate polymer compound coated Fe3O4 nanocomposites. The nanometer oil displacement agent no.3 is composed of water, oil, surfactants, isopropanol, and sodium chloride; The surfactant is a combination of polyoxyethylene ether non ionic surfactants and sulfonated saponins. The nanometer oil displacement agent no.4 is composed of nano silica, diphenyl ether based gemini surfactants, isothiazolinone derivatives, enhancers, stabilizers, dodecane, tetradecane, and water.

Round 3
Reviewer 5 Report
The authors responded appropriately to my observations. The article can be published in the form presented.